# Analysis of Volatile Compounds in Coffee Prepared by Various Brewing and Roasting Methods

**DOI:** 10.3390/foods10061347

**Published:** 2021-06-10

**Authors:** Ja-Myung Yu, Mingi Chu, Hyunbeen Park, Jooyeon Park, Kwang-Geun Lee

**Affiliations:** Department of Food Science and Biotechnology, Dongguk University-Seoul, 32, Dongguk-ro, Ilsandong-gu, Goyang-si 10326, Korea; yujm91@hanmail.net (J.-M.Y.); alsrl0229@naver.com (M.C.); hbeen14@naver.com (H.P.); pjy041100@naver.com (J.P.)

**Keywords:** volatiles, coffee, brewing, roasting, grinding

## Abstract

Volatile compounds of coffee brewed under various roasting conditions and by different brewing methods were analyzed. Green coffee beans (*Coffea arabica*) were roasted at 235 °C for 13 min, 240 °C for 15 min, and 245 °C for 17 min. Roasted coffee beans were ground into particles of three different sizes (710, 500, and 355 μm) and brewed by an espresso coffee machine and the cold brew method. Three types of water (filtered, tap, and bottled) were used for coffee extraction. SPME-GC-MS results indicated that increasing the roasting temperature and time increased the levels of 2,2′-methylene-bis-furan, guaiacol, and 4-ethylguaiacol (*p <* 0.05) and decreased the levels of furfural (*p <* 0.05). Grind size was inversely proportional to the measured signal of volatiles by GC-MS (*p <* 0.05). The measured GC/MS intensities of 2-methylpyrazine, 2,5-dimethylpyrazine, and 2-methoxy-4-vinylphenol were significantly higher in coffee brewed with filtered water (*p* < 0.05) than tap and bottled water. 2-Methylpyrazine, 1-methylpyrrole, and 2-acetylfuran were the most abundant components in the cold brew. Overall, roasting conditions and extraction methods were determined to be significant factors for volatile compounds in coffee. This is the first study showing the analysis of volatile compounds in coffee according to various types of water and extraction methods, such as espresso and cold brew coffee.

## 1. Introduction

Coffee is one of the most consumed beverages around the world. The conventional processing of coffee includes roasting, grinding, and extraction. During roasting, a wide range of chemical reactions occur, including the Maillard reaction, caramelization, and Strecker degradation [1]. These reactions affect not only the color of coffee but also the flavor and aroma [2,3,4,5]. Volatile flavor compounds in coffee are produced from a variety of compounds in green coffee beans, such as reducing sugars, amino acids, lipid, chlorogenic acid, and trigonelline [6].

The chemicals in coffee can be affected by various factors such as roasting conditions, ground particle size, and brewing methods. The grinding process is one of the most important steps for brewing coffee [7,8,9]. The particle size of ground coffee beans plays a significant role in controlling the degree of extraction via the particle’s extraction kinetics [10,11]

Water (including its ionic composition) is an essential ingredient that can highlight the specificities of a coffee or leave it dull and flat [12]. The levels of ions and species in the water change the rate at which water is passed through the ground coffee [12]. It is well known that water treatment is required to remove possible off-flavors deriving from the disinfection (chlorination) process and to prevent limescale deposition associated with the water hardness [13].

Many brewing techniques may be used to prepare coffee [14,15]. *Espresso* is made by forcing hot water at high pressure (maximum of 19 bar) through finely ground coffee. This original Italian-style coffee is increasingly popular in many other countries [16,17]. Differences in flavor characteristics of *espresso* coffees as a result of the coffee variety and extraction temperature have been observed [18].

Cold brew coffee, which has been gaining popularity recently, is known to be smoother and sweeter than conventionally brewed coffee because the flavors and bioactive compounds in coffee are different from the hot water extraction method, such as expresso. Angeloni et al. reported that significant differences were found in the physicochemical parameters and sensory evaluation. Cold brew coffee was recognized as being less bitter with more contents of caffeine and chlorogenic compounds than expresso since higher temperature gives rise to an increase of solid compounds such as caffeine [19]. According to market research, the global cold brew market size was valued at USD 339.7 million in 2018 and is expected to reach USD 1.63 billion by 2025 [20]. Further, aside from its more appealing and less-acidic taste, cold brew coffee can contain up to two-fold more caffeine than hot brew coffee [21]. In addition, crude polysaccharides isolated from cold brew coffee serve as an inducer of the systemic immune system through the intestinal immune system [22]. Despite the growing popularity of cold brew coffee, there is little published on the chemistry or associated benefits or health risks of cold brew coffee.

Despite the growing popularity of cold brew coffee, there is little published on the chemistry or associated benefits or health risks of cold brew coffee. In this study, the volatile compounds produced in coffee under various roasting and brewing conditions, such as the grind size, types of water, and extraction methods, were analyzed and compared.

## 2. Materials and Methods

### 2.1. Chemical Reagents and Materials

Green coffee beans (*Coffea arabica*, from Brazil) were bought from a commercial market in Seoul, Korea. The origin of green coffee beans was verified by the importers and experts. 2-Methylpyrazine, 2,5-dimethylpyrazine, 2,6-dimethylpyrazine, 2-ethyl-3-methylpyrazine, 2-ethyl-5-methylpyrazine, 3-ethyl-2,5-dimethylpyrazine, furfural, furfuryl acetate, furfuryl propionate, furfuryl alcohol, 1-furfurylpyrrole, 4-ethylguaiacol, 2-methoxy-4-vinylphenol, 5-methylfurfural, quinoxaline, and C7–C30 alkane standard were obtained from Sigma–Aldrich Chemical Co. (St. Louis, MO, USA). 2-Ethylpyrazine, 2,3-dimethylpyrazine, 1-methylpyrrole, guaiacol, and 2-acetylfuran were bought from Tokyo Chemical Industry Co., Ltd. (Tokyo, Japan). Divinylbenzene/carboxen/polydimethylsiloxane (DVB/CAR/PDMS, 50-μm film thickness) solid-phase microextraction (SPME) fiber was purchased from Supelco, Inc. (Bellefonte, PA, USA).

### 2.2. Sample Preparation

Green coffee beans were roasted under three different conditions (235 °C for 13 min, 240 °C for 15 min, and 245 °C for 17 min) in a coffee bean roaster (CBR-101A, Gene Café, Korea). Each roasted bean was ground in a grinder (Hanil, Seoul, Korea) for 30 s twice. These samples passed through testing sieves (710, 500, and 355 μm) for 2 min twice in a sieve shaker. Two different extraction methods (expresso and cold brew) were then applied. For espresso coffee, roasted ground coffee (12.5 g) was brewed with 100 mL of water using an espresso coffee machine (BCC-480ES, Bean Cruise, Korea). Cold brew coffee was made based on the New York Times’ cooking website. A sample of 12.5 g was placed in 100 mL of water at room temperature for 4 h, and then at 4 °C for 8 h [21]. Three types of water (filtered, tap, and bottled water) were used for coffee extraction.

In total, fifty-four coffee samples were prepared, and the sample names were abbreviated from E-1 to E-27 (extraction method), and from C-1 to C-27 (cold brew). According to the roasting conditions, sample numbers were divided into 1 to 9 (235 °C for 13 min), 10–18 (240 °C for 15 min), and 19–27 (245 °C for 17 min). The sample number was also assigned to the particle size of grounded coffee (355 μm: 1–3, 10–12, 19–21; 500 μm: 4–6, 13–15, 22–24; 710 μm: 7–9, 16–18, 25–27). The sample number assigned according to the size of the coffee bean was divided into three equal parts according to the type of water (filtered water: 1, 4, 7, 10, 13, 16, 19, 22, 25; tap water: 2, 5, 8, 11, 14, 17, 20, 23, 26; bottled water: 3, 6, 9, 12, 15, 18, 21, 24, 27).

### 2.3. Analysis of Volatile Compounds

Volatile compounds in coffee were extracted using SPME. A 10 mL of sample was added to the GC vial. The internal standard quinoxaline (10 μL), the alkane standard (20 μL), and a magnetic stirring bar were added. After stirring the samples at 70 °C for 10 min, to reach equilibrium, the SPME fiber was injected into the vial at 70 °C for 40 min for adsorption of volatile compounds. Afterward, the fiber was inserted into the gas chromatograph. Gas chromatography–mass spectrometry (GC-MS) analysis was performed using a DB-WAX column (length × inside diameter × phase thickness: 60 m × 250 μm × 0.25 μm) by modifying an existing method [6]. Helium was used as the carrier gas, with a flow rate of 1.0 mL/min and splitless mode (splitless time: 1 min). The oven temperature was maintained at 44 °C for 5 min, increased to 170 °C at 3 °C/min and held for 10 min, and then raised to 240 °C at 8 °C/min and held for 5 min. Volatile compounds were identified by their retention index (RI), co-injection, and by comparison of their mass spectra with those published in the Wiley mass spectrum database. The peak area ratio (peak area of each peak/peak area of internal standard) of each compound was calculated from the peak area of the internal standard. All analyses were conducted in three replications, and average value and standard deviations were calculated.

### 2.4. Statistical Analysis

Statistical analysis was performed with IBM SPSS Statistics 23 (IBM, Armonk, NY, USA), and graphs were constructed with GraphPad Prism 5.0 software (GraphPad Software, Inc., San Diego, CA, USA). All data were analyzed by one-way ANOVA and Duncan’s multiple range test for investigating significant differences (*p* < 0.05). Principal component analysis (PCA) was performed on the mean values of peak area ratio using XLSTAT (version 2018; Addinsoft, Paris, France).

## 3. Results

### 3.1. Analysis of Volatile Compounds in Coffee

The selected 24 volatile compounds are shown in Table 1. These selected volatile compounds are the main volatiles identified in coffee samples. In addition, these 24 volatiles were analyzed the most in our study and represent the aroma’s functional groups (chemical types) in coffee. Identification of the volatile compounds was based on their RI, co-injection, and comparison of their mass spectrum with those in the Wiley mass spectrum database. All values are represented as the peak area ratio (peak area of each peak/peak area of internal standard). Table 2 shows the peak area ratio of the volatile compounds in E-1 to E-9 (espresso roasted at 235 °C for 13 min) and C-1 to C-9 (cold brew roasted at 235 °C for 13 min). Among the espresso coffees, the finest grinds (355 μm) produced significantly higher measured GC/MS intensities of all 24 volatile compounds, except for 2,2′-[oxybis (methylene)]bis-furan and 2-methoxy-4-vinylphenol, compared with the medium (500 μm) and large (750 μm) grind sizes (*p* < 0.05). For espresso coffees extracted with different types of water, bottled water and tap water led to high measured GC/MS intensities of all 24 volatile compounds. Comparing cold brew coffees, fine grinds (355 μm) resulted in significantly high measured GC/MS intensities of volatile compounds, respectively (*p* < 0.05).

For cold brew coffees (C-1 to C-9), filtered water led to significantly higher measured GC/MS intensities of volatile compounds (*p <* 0.05). The water used in this study has different amounts of carbonates and bicarbonates. The levels of carbonates and bicarbonate were much lower in the filtered water compared to the bottled and tap water. A study by Gardner (1958) showed that the carbonates and bicarbonates of sodium ions slowed down the brewing time. Therefore, cold brew, which is extracted by soaking coffee beans in water, appears to have been more affected by the extraction of volatile compounds than *espresso* coffee [12].

Table 3 shows the volatile compounds in both types of coffee roasted at 240 °C for 15 min. Among the espresso coffees (E-10 to E-18), the smaller particle of ground coffee (355 μm) corresponded to significantly higher measured GC/MS intensities of volatile compounds (*p* < 0.05). Among the espresso coffees extracted with different types of water (E-10, E-11, and E-12), filtered water resulted in significantly higher measured GC/MS intensities of 18 volatile compounds (*p* < 0.05). Comparing E-13, E-14, and E-15, the measured GC/MS intensities of 22 volatile compounds, excluding 2-methylpyrazine and furfuryl alcohol, were notably high when filtered water was used (*p* < 0.05). Comparing E-16, E-17, and E-18, tap water led to significantly higher measured GC/MS intensities of 21 volatile compounds (*p* < 0.05). Comparing cold brew samples, fine grinds (355 μm) occasioned notably high levels of volatile compounds, respectively (*p* < 0.05). Among C-10 to C-12, the measured GC/MS intensities of all 24 volatile compounds were similarly detected. Among C-13 to C-15 and C-16 to C-18, filtered water gave rise to remarkably high measured GC/MS intensities of 19 and 2 volatile compounds, respectively (*p* < 0.05). For extraction with filtered water, more volatiles could be expected to be detected as grind size decreases and roasting temperature increases. However, in this study, espresso prepared from filtered water and at a relatively low roasting temperature did not generate a high concentration of volatiles. Similarly, cold brew prepared from fine grinds and filtered water did not have a high concentration of volatiles. The reason for this result should be examined in future studies.

Table 4 lists the volatile compounds in both types of coffee roasted at 245 °C for 17 min (E-19 to E-27 and C-19 to C-27). Comparing expresso coffee samples, fine grinds (355 μm) led to significantly high measured GC/MS intensities of volatile compounds, respectively (*p* < 0.05). Among E-19 to E-21, filtered water led to slightly but not significantly high measured GC/MS intensities of all 24 volatile compounds (*p* > 0.05), except for isopropenyl pyrazine, 2,6-diethylpyrazine, and 2-methoxy-4-vinylphenol. Among E-22 to E-24 and E-25 to E-27, filtered water resulted in significantly high measured GC/MS intensities of 6 and 18 volatile compounds, respectively (*p* < 0.05). Comparing cold brew samples, there were no or highly significant differences in grind sizes 500 and 710 μm. Filtered water occasioned significantly high measured GC/MS intensities of 1-furfurylpyrrole and 2-methoxy-4-vinylphenol among cold brews C-19 to C-21 (*p* < 0.05), furfuryl alcohol among cold brews C-22 to C-24 (*p* < 0.05), and 12 volatile compounds among cold brews C-25 to C-27 (*p* < 0.05). Comparing volatile compounds of middle size (500 μm) of ground coffee, the levels of 2-methylpyrazine, 2,5-Dimethylpyrazine, 2,6-Dimethylpyrazine, 2-ethylpyrazine, 2-ethyl-5-methylpyrazine, 3-ethyl-2,5-dimethylpyrazine, 2-acetylfuran, 2,2′-Bifuran, 5-Methylfurfural, and 1-methylpyrrole in cold brew coffee were higher than espresso coffee. These flavor compounds have a sweet, nutty, and fruity odor [23,24,25,26].

In Figure 1, among the many volatile compounds analyzed, significantly increased or decreased compounds were selected to show their levels. As shown in Figure 1a, increasing the roasting temperature and time increased the levels of 2,2′-methylene-bis-furan, guaiacol, and 4-ethylguaiacol (*p <* 0.05), and decreased the levels of furfural and 5-methylfurfural (*p <* 0.05). In Moon and Shibamoto’s (2009) study, intensifying the roasting conditions of green coffee beans yielded relatively lower furanone derivatives and furfural. This result is possibly associated with the interconversion of furan, furfural, furfuryl alcohol, and 2-methylfuran. In particular, the high activation energy of furfural reduction to furan is shown to be thermodynamically favored [27]. In addition, the formation of aroma compounds, such as pyrroles, generated by the Maillard reaction between reducing sugars and amino acids, were formed more in coffee beans roasted under higher roasting conditions than mild roasting conditions [3].

As shown in Figure 1b, the smaller the grind size, the higher the measured GC/MS intensities of 2-ethyl-5-methylpyrazine, 4-ethylguaiacol, and 2-acetylfuran (*p* < 0.05). As the grind size decreases, the solid–liquid interfacial area increases, in turn, increasing the levels of volatile compounds extracted [11]. Other studies reported that coffee prepared with rough coffee grounds had the lowest aromatic profile [16,28]. Trigonelline, chlorogenic acids, and lipids increased inversely with grind size. The caffeine content also increased significantly as grind size decreased [18,29]. These results suggest that different aromatic profiles of coffee can be obtained when different grind sizes are used. As the size of the particles decreases, the packing effect becomes higher, and the perfusion of the water in coffee samples becomes difficult. Therefore, the particle size is usually optimized according to the type of extraction methods, such as the machine used for the preparation of the coffee. In a future study, the optimization of coffee machines could be carried out.

In Figure 1c, high measured GC/MS intensities of 2-methylpyrazine, 2,5-dimethylpyrazine, and 2-methoxy-4-vinylphenol were observed in coffee brewed with filtered water than bottled water. Water quality can have a direct impact on the quality of espresso coffee. Brewed coffees often differed in flavor and appearance, depending on whether the water was distilled, soft, or hard. Beverages made with solutions containing carbonates were the least desirable, having a flat and dull characteristic [12].

### 3.2. Principal Component Analysis (PCA) for Volatile Compounds in a Coffee Model System

It is difficult to determine the difference in the mean of the volatile compounds produced by roasting and extraction methods because of the complex dynamics of the system. PCA is an effective mathematical method to reduce the dimensions of multivariate data [30]. The character with a number on the PCA biplot represents the volatile compounds defined in Table 1. PCA can identify where coffee samples made under different conditions, such as roasting conditions, grind coffee bean size, brewing methods, and types of water, are located on a volatile compounds data map [31].

Figure 2a shows the PCA results explaining the relative location of volatile compounds in espresso coffee. The PCA biplot explains approximately 79.75% of the variability. PC1 explains 52.81% of the total variability as the primary axis, and PC2 explains 26.95% of the entire variability as the vertical axis. One of the drawbacks of a PCA biplot is that it only explains a certain percentage of the total variability; that is, not all of the dataset is reflected in the PCA biplot [32]. Espresso coffees E-1 to E-9 (roasted at 235 °C for 13 min) were located in the second quadrant. The first group consisted of E-2 and E-3, which contained high amounts of 2-ethyl-5-methylpyrazine, 2,2′-bifuran, furfural, 5-methylfurfural, and 2-methoxy-4-vinylphenol (a8, b2, b8, b9, d3). E-10 to E-27 (coffee roasted at 240 °C for 15 min and 245 °C for 17 min) were located in the negative PC2. The second group consisted of E-19, E-20, and E-21, associated with high levels of 2,2′-methylene-bis-furan, 2,2′-[oxybis(methylene)]bis-furan, 1-methylpyrrole, guaiacol, and 4-ethylguaiacol (b3, b4, c1 d1, d2). The 355-μm (E-1 to E-3, E-10 to E-12, and E-19 to E-21) and 710-μm grind sizes (E-7 to E-9, E-16 to E-18, E-25 to E-27) were located in positive PC1 and negative PC1, respectively.

Figure 2b shows the PCA results and describes the relative location of volatile compounds in cold brew coffee. The PCA biplot explains about 78.82% of the variability. Most of the variability, 51.63%, was attributed to PC1, with PC2 (the vertical axis) accounting for just 27.19% of the total variability. Cold brew coffees C-2 to C-9, which were roasted at 235 °C for 13 min, were located in the fourth quadrant. The first group included C-2 and C-4, with high levels of 2,6-diethylpyrazine, furfural, 5-methylfurfural, and 2-methoxy-4-vinylphenol (a6, b8, b9, d3). The second group consisted of C-14 to C-27, which contained high levels of 2,2′-methylene-bis-furan, 2,2′-[oxybis(methylene)]bis-furan, guaiacol, 4-ethylguaiacol, and 1-methylpyrrole (b3, b4, c1, d1, d2).

Figure 2 shows the PCA results and explains the relative location of volatile compounds in espresso and cold brew coffee roasted at 240 °C for 15 min. Together, PCI (60.62%) and PC2 (15.53%) describe 76.14% of the total variability. Espresso coffees (E-10 to E-18) had positive PC2 scores. Cold brew coffees (C-10 to C-18) had negative PC2 scores. Espresso coffee contained high levels of 2,3-dimethylpyrazine, 2,2′-methylene-bis-furan, 2,2′-[oxybis(methylene)]bis-furan, 1-furfurylpyrrole, guaiacol, 4-ethylguaiacol, and 2-methoxy-4-vinylphenol (a2, b3, b4, c2, d1, d2, d3). Conversely, cold brew contained high levels of 2-methylpyrazine, 3-ethyl-2,5-dimethylpyrazine, furfural, 5-methylfurfural, and 1-methylpyrrole (a1, a9, b8, b9, c1).

## 4. Discussion

In this study, the higher the roasting temperature and the longer the roasting time, the higher the concentrations of 2,2′-methylene-bis-furan, guaiacol, and 4-ethylguaiacol and the lower the concentration of furfural. Volatile compounds from green coffee beans were reduced as the intensity of the roasting conditions increased. On the contrary, aroma chemicals produced by the Maillard reaction were increased under higher roasting conditions than mild roasting conditions. The smaller the grind size, the higher the concentration of volatile compounds. As the grind size decreases, the solid–liquid interfacial area increases, in turn, increasing the amount of volatiles that can be extracted. Types of water with different ions can have an impact on the flavor and aroma of coffee. 2-Methylpyrazine, 1-methylpyrrole, and 2-acetylfuran were the most abundant components in the cold brew coffee. Overall, the roasting conditions and extraction methods were determined to be significant factors for volatile compounds in coffee.

## Figures and Tables

**Figure 1 foods-10-01347-f001:**
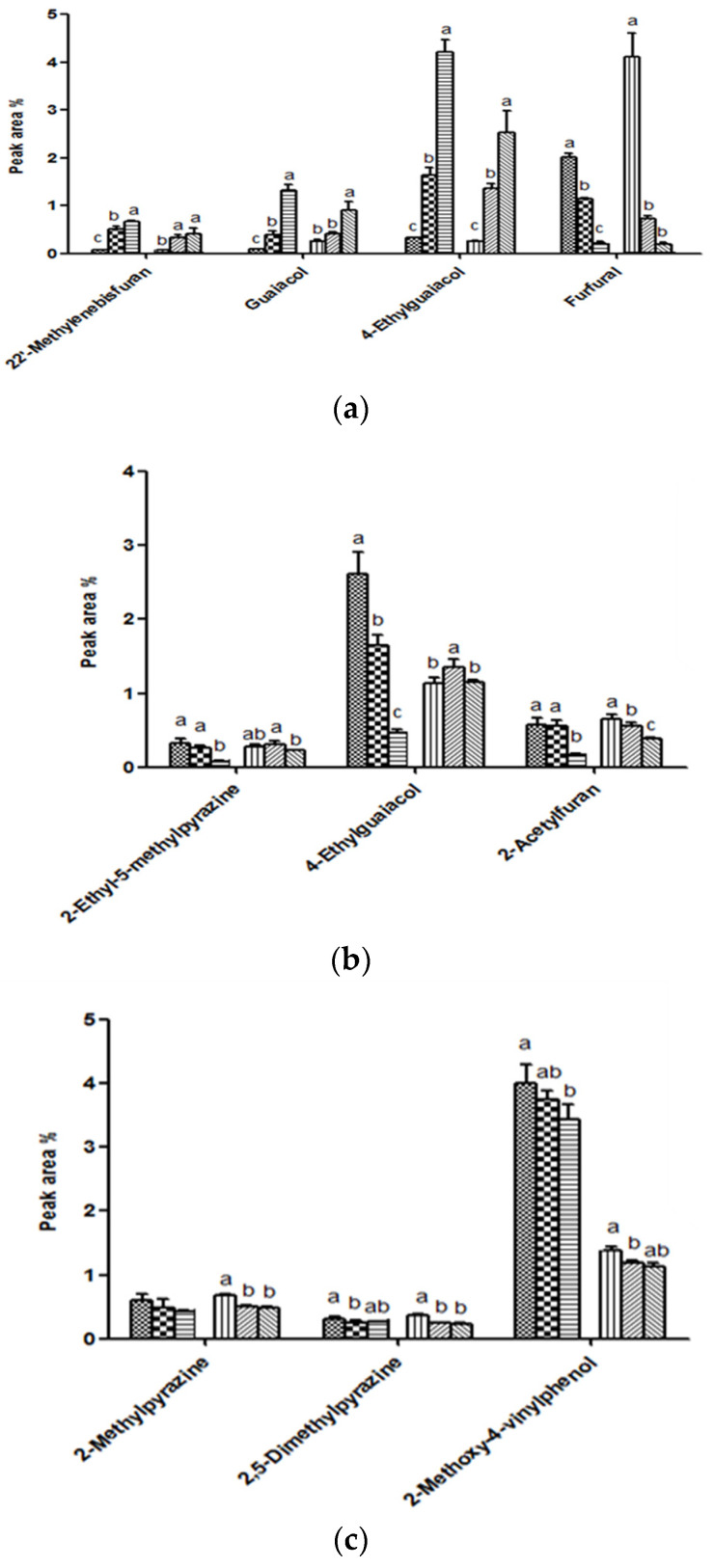
Levels of volatile compounds in coffee according to (**a**) roasting time and temperature (E-4 (
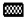
) and C-4 (
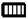
): roasted at 235 °C for 13 min; E-13 (
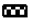
) and C-13 (
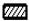
): roasted at 240 °C for 15 min; E-22 (
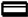
) and C-22 (
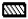
): roasted at 245 °C for 17 min); (**b**) grind size (E-10 (
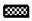
) and C-10 (
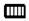
): 355 μm; E-13 (
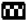
) and C-13 (
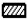
): 500 μm; E-16 (
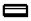
) and C-16 (
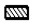
): 710 μm); (**c**) types of water (E-13 (
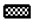
) and C-13 (
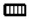
): filtered water; E-14 (
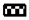
) and C-14 (
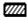
): tap water; E-15 (
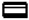
) and C-15 (
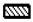
): bottled water). Different letters (a, b, c) represented significant differences.

**Figure 2 foods-10-01347-f002:**
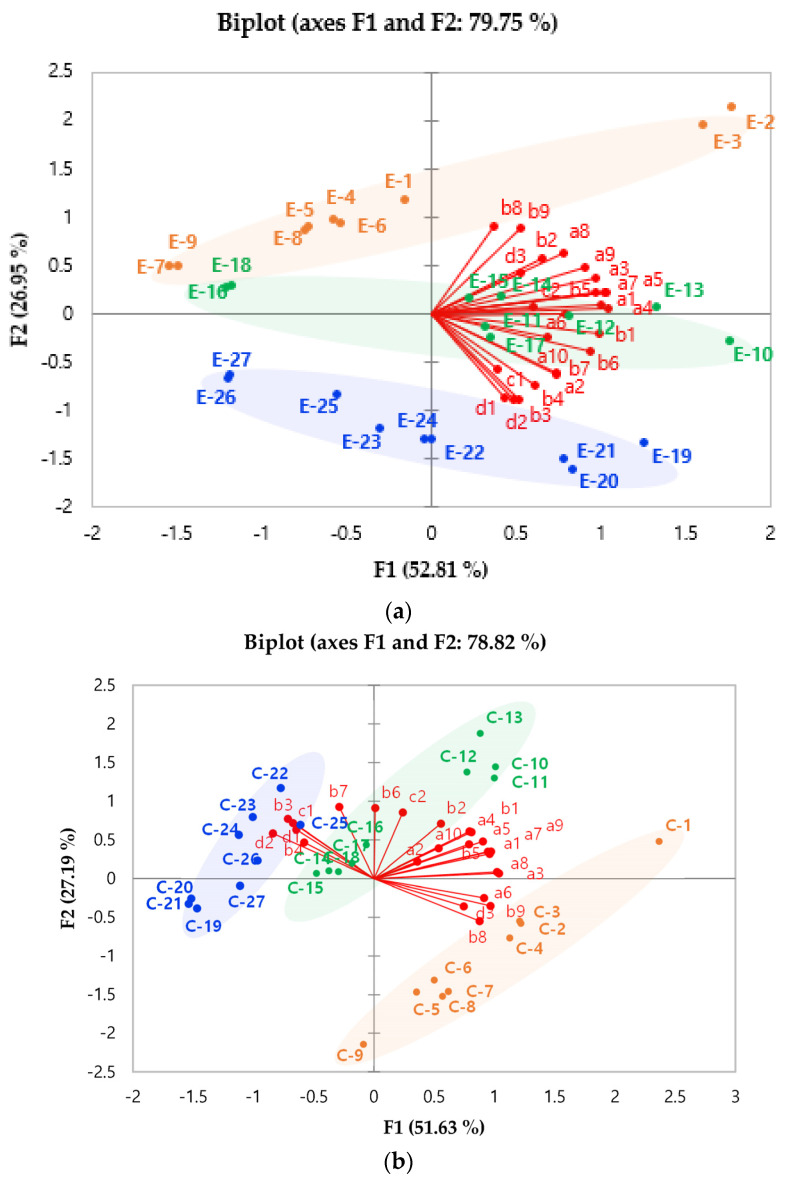
Principal component analysis (PCA) biplot of volatile compounds (**a**) in espresso coffee, (**b**) cold brew coffee. The relative location of volatile compounds in espresso and cold brew coffee roasted at 240 °C for 15 min. (a1: 2-methylpyrazine, a2: 2,3-dimethylpyrazine, a3: 2,5-dimethylpyrazine, a4: 2,6-dimethylpyrazine, a5: 2-ethylpyrazine, a6: 2,6-diethylpyrazine, a7: 2-ethyl-3-methylpyrazine, a8: 2-ethyl-5-methylpyrazine, a9: 3-ethyl-2,5-dimethylpyrazine, a10: isopropenyl pyrazine, b1: 2-acetylfuran, b2: 2,2′-bifuran, b3: 2,2′-methylenebisfuran, b4: 2,2′-[oxybis(methylene)]bisfuran, b5: furfuryl alcohol, b6: furfuryl acetate, b7: furfuryl propionate, b8: furfural, b9: 5-methylfurfural, c1: 1-methylpyrrole, c2: 1-furfurylpyrrole, d1: guaiacol, d2: 4-ethylguaiacol, d3: 2-methoxy-4-vinylphenol.). E and C mean extraction method of expresso and cold brew, respectively. According to the roasting conditions, sample numbers were divided into 1 to 9 (235 °C for 13 min), 10–18 (240 °C for 15 min), and 19–27 (245 °C for 17 min). The sample number was also assigned to the particle size of grounded coffee (355 μm: 1–3, 10–12, 19–21; 500 μm: 4–6, 13–15, 22–24; 710 μm: 7–9, 16–18, 25–27). The sample number assigned according to the size of the coffee bean was divided into three equal parts according to the type of water (filtered water: 1, 4, 7, 10, 13, 16, 19, 22, 25; tap water: 2, 5, 8, 11, 14, 17, 20, 23, 26; bottled water: 3, 6, 9, 12, 15, 18, 21, 24, 27).

**Table 1 foods-10-01347-t001:** Volatile compounds in coffee (E-1: espresso brewing method, roasted at 235 °C for 13 min, particle size of ground coffee bean 355 μm, and purified water).

No.	Compounds	Retention Time	R.I	R.I (Ref)	Co- Injection	Mass Spectrum
a1	2-Methylpyrazine	25.99	1289	1288	o	o
a2	2,3-Dimethylpyrazine	29.87	1370	1371	o	o
a3	2,5-Dimethylpyrazine	28.73	1346	1348	o	o
a4	2,6-Dimethylpyrazine	29.02	1352	1354	o	o
a5	2-Ethylpyrazine	29.23	1356	1359	o	o
a6	2,6-Diethylpyrazine	33.83	1456	1444		o
a7	2-Ethyl-3-methylpyrazine	32.54	1427	1422	o	o
a8	2-Ethyl-5-methylpyrazine	31.64	1407	1415		o
a9	3-Ethyl-2,5-dimethylpyrazine	34.32	1467	1452	o	o
a10	Isopropenyl pyrazine	40.82	1619			o
b1	2-Acetylfuran	36.74	1522	1527	o	o
b2	2,2′-Bifuran	40.47	1610	1614		o
b3	2,2′-Methylenebisfuran	40.93	1622	1615		o
b4	2,2′-[Oxybis(methylene)]bisfuran	57.71	1997	1986		o
b5	Furfuryl alcohol	42.84	1669	1666	o	o
b6	Furfuryl acetate	37.82	1547	1552	o	o
b7	Furfuryl propionate	40.36	1607	1603	o	o
b8	Furfural	34.81	1477	1482	o	o
b9	5-Methylfurfural	39.63	1590	1596	o	o
c1	1-Methylpyrrole	19.30	1147	1140	o	o
c2	1-Furfurylpyrrole	49.66	1841	1833	o	o
d1	Guaiacol (2-Methoxyphenol)	51.10	1874	1872	o	o
d2	4-Ethylguaiacol	59.93	2052	2054	o	o
d3	2-Methoxy-4-vinylphenol	64.80		2156	o	o

Identification for volatile compounds compared with Retention Index (RI) on DB-WAX column in NIST and VCF library, co-injection, and mass spectrum.

**Table 2 foods-10-01347-t002:** Volatile compounds (PAR: peak area ratio) in coffee roasted at 235 °C for 13 min (model E-1 to E-9 for expresso, C-1 to C-9 for cold brew). The sample number was also assigned to the particle size of grounded coffee (355 μm: 1–3; 500 μm: 4–6; 710 μm: 7–9). The sample number assigned according to the size of the coffee bean was divided into three equal parts according to the type of water (filtered water: 1, 4, 7; tap water: 2, 5, 8; bottled water: 3, 6, 9).

Compounds	235 °C, 13 min																	
	355 μm						500 μm						710 μm					
	Filtered Water		Tap Water		Bottled Water		Filtered Water		Tap Water		Bottled Water		Filtered Water		Tap Water		Bottled Water	
	E-1	C-1	E-2	C-2	E-3	C-3	E-4	C-4	E-5	C-5	E-6	C-6	E-7	C-7	E-8	C-8	E-9	C-9
2-Methylpyrazine	0.382 ^b^	0.885 ^a^	0.941 ^a^	0.615 ^b^	0.852 ^a^	0.596 ^b^	0.338	0.641 ^a^	0.288	0.505 ^b^	0.324	0.520 ^b^	0.190 ^b^	0.605 ^a^	0.436 ^a^	0.591 ^a^	0.186 ^b^	0.493 ^b^
2,3-Dimethylpyrazine	0.048 ^b^	0.070 ^b^	0.068 ^ab^	0.087 ^ab^	0.056 ^a^	0.098 ^a^	0.040	0.097 ^a^	0.031	0.075 ^b^	0.041	0.056 ^c^	0.019 ^b^	0.056 ^a^	0.032 ^a^	0.046 ^b^	0.022 ^b^	0.039 ^b^
2,5-Dimethylpyrazine	0.237 ^b^	0.523 ^a^	0.553 ^a^	0.371 ^b^	0.518 ^a^	0.346 ^b^	0.213	0.383 ^a^	0.188	0.304 ^b^	0.205	0.302 ^b^	0.104 ^b^	0.352 ^a^	0.224 ^a^	0.340 ^a^	0.102 ^b^	0.269 ^b^
2,6-Dimethylpyrazine	0.215 ^b^	0.438 ^a^	0.451 ^a^	0.292 ^b^	0.435 ^a^	0.298 ^b^	0.184	0.302	0.164	0.245	0.182	0.253	0.090 ^b^	0.279 ^a^	0.187 ^a^	0.266 ^ab^	0.093 ^b^	0.213 ^b^
2-Ethylpyrazine	0.189 ^b^	0.338 ^a^	0.372 ^a^	0.252 ^b^	0.366 ^a^	0.243 ^b^	0.161	0.258 ^a^	0.143	0.197 ^b^	0.153	0.206 ^b^	0.072 ^b^	0.218	0.148 ^a^	0.209	0.074 ^b^	0.174
2,6-Diethylpyrazine	0.082 ^b^	0.219	0.100 ^ab^	0.213	0.125 ^a^	0.209	0.057	0.199 ^a^	0.052	0.163 ^b^	0.061	0.172 ^a^	0.032	0.114 ^a^	0.035	0.104 ^a^	0.044	0.079 ^b^
2-Ethyl- 3-methylpyrazine	0.247 ^b^	0.402	0.489 ^a^	0.312	0.482 ^a^	0.316	0.211	0.312 ^a^	0.196	0.245^b^	0.219	0.263 ^b^	0.096 ^b^	0.297 ^a^	0.189 ^a^	0.297 ^a^	0.099 ^b^	0.231 ^b^
2-Ethyl- 5-methylpyrazine	0.317 ^b^	0.326 ^a^	0.384 ^ab^	0.256 ^b^	0.393 ^a^	0.278 ^ab^	0.262	0.259	0.213	0.217	0.225	0.217	0.104	0.237 ^ab^	0.141	0.257 ^a^	0.140	0.205 ^b^
3-Ethyl- 2,5-imethylpyrazine	0.422 ^b^	0.753 ^a^	1.059 ^a^	0.587 ^b^	1.027 ^a^	0.613 ^ab^	0.336 ^a^	0.588	0.297 ^b^	0.491	0.347 ^a^	0.522	0.139 ^b^	0.500 ^a^	0.398 ^a^	0.462 ^a^	0.145 ^b^	0.399 ^b^
Isopropenyl pyrazine	0.048 ^b^	0.104	0.120 ^a^	0.100	0.115 ^a^	0.100	0.043	0.098	0.047	0.093	0.052	0.098	0.026 ^b^	0.106 ^a^	0.038 ^a^	0.105 ^a^	0.019 ^b^	0.082 ^b^
2-Acetylfuran	0.308 ^b^	0.738 ^a^	0.643 ^a^	0.551 ^b^	0.632 ^a^	0.541 ^b^	0.239	0.475	0.229	0.380	0.264	0.392	0.133 ^b^	0.382 ^a^	0.238 ^a^	0.354 ^ab^	0.147 ^b^	0.260 ^b^
2,2′-Bifuran	0.086 ^b^	0.088	0.267 ^a^	0.092	0.222 ^a^	0.092	0.069 ^ab^	0.077 ^a^	0.062 ^b^	0.060 ^b^	0.077 ^a^	0.067 ^ab^	0.045 ^b^	0.056 ^ab^	0.127 ^a^	0.062 ^a^	0.061 ^b^	0.045 ^b^
2,2′-Methylenebisfuran	0.078 ^b^	0.086	0.225 ^a^	0.086	0.193 ^a^	0.092	0.072 ^b^	0.052	0.071 ^b^	0.043	0.081^a^	0.046	0.047 ^b^	0.041 ^a^	0.103 ^a^	0.041 ^a^	0.054 ^b^	0.031 ^b^
2,2′-[Oxybis (methylene)]bisfuran	0.124 ^b^	0.235	0.591 ^a^	0.238	0.612 ^a^	0.259	0.117 ^a^	0.811 ^b^	0.087 ^b^	0.964 ^ab^	0.112^ab^	1.063 ^a^	0.081 ^c^	0.247	0.244 ^a^	0.226	0.154 ^b^	0.215
Furfuryl alcohol	1.656 ^b^	2.758 ^a^	2.710 ^a^	1.967 ^b^	2.514 ^a^	1.813 ^b^	1.405	1.729 ^ab^	1.335	1.495 ^b^	1.355	1.788 ^a^	0.728 ^b^	1.479	1.379 ^a^	1.553	0.768 ^b^	1.349
Furfuryl acetate	0.891 ^b^	1.916	2.194 ^a^	1.545	2.321 ^a^	1.532	0.706 ^ab^	1.190 ^a^	0.636 ^b^	0.832 ^b^	0.731^a^	0.937 ^b^	0.289 ^b^	0.749	0.715 ^a^	0.775	0.332 ^b^	0.627
Furfuryl propionate	0.058 ^b^	0.103	0.091 ^a^	0.083	0.103 ^a^	0.084	0.043 ^ab^	0.055	0.039 ^b^	0.045	0.050^a^	0.045	0.013 ^b^	0.037 ^a^	0.033 ^a^	0.036 ^a^	0.010 ^b^	0.028^b^
Furfural	2.604 ^b^	4.906 ^a^	5.492 ^a^	3.746 ^b^	4.828 ^a^	3.665 ^b^	2.015	4.096 ^a^	1.850	3.236 ^b^	1.990	3.381 ^ab^	1.196 ^b^	3.500	2.338 ^a^	3.577	1.124 ^b^	2.823
5-Methylfurfural	2.716 ^b^	6.897 ^a^	7.229 ^a^	5.392 ^b^	6.608 ^a^	5.268 ^b^	2.236	4.682 ^a^	2.051	3.700 ^b^	2.210	4.190 ^ab^	1.176 ^b^	3.503	2.429 ^a^	3.559	1.168 ^b^	3.106
1-Methylpyrrole	0.076 ^b^	0.167	0.114 ^a^	0.134	0.147 ^a^	0.128	0.055	0.156 ^a^	0.056	0.113 ^b^	0.064	0.121 ^b^	0.024 ^b^	0.133 ^ab^	0.075 ^a^	0.140 ^a^	0.025 ^b^	0.103 ^b^
1-Furfurylpyrrole	1.548 ^a^	0.761 ^a^	1.204 ^b^	0.612 ^b^	1.087 ^b^	0.643 ^ab^	1.062	0.565	1.210	0.471	1.258	0.495	0.599	0.395	0.490	0.401	0.510	0.357
Guaiacol	0.108 ^b^	0.352 ^a^	0.184 ^a^	0.248 ^b^	0.165 ^a^	0.249 ^b^	0.090	0.257	0.083	0.232	0.089	0.229	0.055	0.055	0.064	0.051	0.056	0.045
4-Ethylguaiacol	0.337 ^b^	0.366	0.976 ^a^	0.336	0.944 ^a^	0.357	0.334	0.242 ^b^	0.321	0.242 ^b^	0.344	0.340 ^a^	0.190 ^b^	0.444	0.404 ^a^	0.456	0.168 ^b^	0.429
2-Methoxy- 4-vinylphenol	3.612 ^a^	1.367	2.937 ^b^	1.388	2.851 ^b^	1.412	3.253 ^b^	1.520 ^b^	3.348 ^b^	1.576 ^b^	3.735^a^	1.727 ^a^	1.647 ^a^	1.311 ^b^	1.469 ^b^	1.431 ^a^	1.286 ^c^	1.477 ^a^

All values represented as the average of three replicates are the peak area ratio (peak area of each peak/peak area of internal standard). Different letters of the alphabet paired with a cardinal number within a particle size and extraction method indicate significant differences in the type of water according to Duncan’s test between each sample (*p* < 0.05).

**Table 3 foods-10-01347-t003:** Volatile compounds in coffee roasted at 240 °C for 15 min (model E-10 to E-18 for expresso, C-10 to C-18 for cold brew). The sample number was also assigned to the particle size of grounded coffee (355 μm: 10–12; 500 μm: 13–15; 710 μm: 16–18). The sample number assigned according to the size of the coffee bean was divided into three equal parts according to the type of water (filtered water: 10, 13, 16; tap water: 11, 14, 17; bottled water: 12, 15, 18).

Compounds	240 °C, 15 min																	
	355 μm						500 μm						710 μm					
	Filtered Water		Tap Water		Bottled Water		Filtered Water		Tap Water		Bottled Water		Filtered Water		Tap Water		Bottled Water	
	E-10	C-10	E-11	C-11	E-12	C-12	E-13	C-13	E-14	C-14	E-15	C-15	E-16	C-16	E-17	C-17	E-18	C-18
2-Methylpyrazine	0.712 ^a^	0.727	0.445 ^b^	0.718	0.476 ^b^	0.645	0.601	0.688 ^a^	0.48	0.496 ^b^	0.427	0.477 ^b^	0.245 ^b^	0.514	0.566 ^a^	0.504	0.221 ^b^	0.476
2,3-Dimethylpyrazine	0.072	0.057	0.085	0.076	0.074	0.064	0.114 ^a^	0.081 ^a^	0.072 ^b^	0.042 ^b^	0.072 ^b^	0.044 ^b^	0.023 ^b^	0.046	0.054 ^a^	0.043	0.028 ^b^	0.045
2,5-Dimethylpyrazine	0.391 ^a^	0.375	0.258 ^b^	0.366	0.281 ^b^	0.345	0.313 ^a^	0.363 ^a^	0.250 ^b^	0.242 ^b^	0.258 ^ab^	0.237 ^b^	0.119 ^b^	0.254	0.279 ^a^	0.249	0.128 ^b^	0.241
2,6-Dimethylpyrazine	0.393 ^a^	0.349	0.272 ^b^	0.350	0.283 ^b^	0.327	0.324 ^a^	0.382 ^a^	0.284 ^a^	0.277 ^b^	0.240 ^b^	0.268 ^b^	0.109 ^b^	0.267	0.273 ^a^	0.263	0.119 ^b^	0.256
2-Ethylpyrazine	0.337 ^a^	0.274	0.224 ^b^	0.257	0.226 ^b^	0.242	0.271 ^a^	0.308 ^a^	0.224 ^b^	0.212 ^b^	0.196 ^b^	0.205 ^b^	0.084 ^b^	0.223	0.227 ^a^	0.220	0.089 ^b^	0.209
2,6-Diethylpyrazine	0.106 ^ab^	0.101	0.040 ^b^	0.118	0.140 ^a^	0.104	0.224 ^a^	0.126 ^a^	0.128 ^b^	0.082 ^b^	0.121 ^b^	0.081 ^b^	0.071	0.088	0.056	0.091	0.071	0.084
2-Ethyl- 3-methylpyrazine	0.419 ^a^	0.332	0.309 ^b^	0.347	0.341 ^b^	0.320	0.386 ^a^	0.386 ^a^	0.302 ^b^	0.267 ^b^	0.310 ^b^	0.265 ^b^	0.144 ^b^	0.262	0.270 ^a^	0.259	0.139 ^b^	0.257
2-Ethyl- 5-methylpyrazine	0.327 ^a^	0.280	0.225 ^b^	0.263	0.247 ^b^	0.266	0.261 ^a^	0.307 ^a^	0.208 ^b^	0.193 ^b^	0.192 ^b^	0.192 ^b^	0.078 ^b^	0.227	0.186 ^a^	0.208	0.099 ^b^	0.204
3-Ethyl- 2,5-imethylpyrazine	0.675 ^a^	0.660	0.436 ^b^	0.807	0.516 ^b^	0.674	0.593 ^a^	0.602 ^a^	0.418 ^b^	0.490 ^b^	0.397 ^b^	0.414 ^b^	0.181^c^	0.526	0.379 ^a^	0.519	0.244 ^b^	0.478
Isopropenyl pyrazine	0.137 ^b^	0.143	N.D ^c^	0.095	0.213 ^a^	0.117	0.164 ^a^	0.112 ^a^	0.111 ^b^	0.078 ^b^	0.097 ^b^	0.080 ^b^	0.076	0.106	0.065	0.095	0.065	0.094
2-Acetylfuran	0.577 ^a^	0.655	0.404 ^b^	0.626	0.441 ^b^	0.587	0.557 ^a^	0.553 ^a^	0.365 ^b^	0.361 ^b^	0.382 ^b^	0.338 ^b^	0.171 ^b^	0.389	0.349 ^a^	0.376	0.170 ^b^	0.375
2,2’-Bifuran	0.214 ^a^	0.107	N.D ^c^	0.096	0.086 ^b^	0.109	0.197 ^a^	0.137	0.116 ^b^	0.065	0.112 ^b^	0.063	0.083 ^b^	0.068	0.177 ^a^	0.069	0.081 ^b^	0.051
2,2’-Methylenebisfuran	0.841 ^a^	0.248	0.486 ^b^	0.224	0.529 ^b^	0.242	0.516 ^a^	0.304	0.356 ^b^	0.262	0.379 ^b^	0.249	0.223 ^b^	0.245	0.699 ^a^	0.247	0.235 ^b^	0.219
2,2’-[Oxybis (methylene)]bisfuran	2.142 ^a^	0.905	1.454 ^b^	0.993	1.601 ^b^	0.988	1.243 ^a^	1.075	1.185 ^ab^	0.939	0.996 ^b^	0.951	0.322 ^b^	0.914	1.274 ^a^	0.920	0.341 ^b^	0.997
Furfuryl alcohol	2.653	2.279	2.333	2.807	2.742	2.888	2.327	2.308 ^a^	2.363	1.923 ^b^	1.985	1.898 ^b^	0.849 ^b^	2.193	1.957 ^a^	2.085	0.825 ^b^	2.114
Furfuryl acetate	4.393 ^a^	2.891	2.364 ^b^	2.677	2.450 ^b^	2.916	2.748 ^a^	3.004	2.075 ^b^	2.628	1.890 ^b^	2.673	0.736 ^b^	2.804	2.567 ^a^	2.405	0.688 ^b^	2.553
Furfuryl propionate	0.354 ^a^	0.197	0.202 ^b^	0.217	0.198 ^b^	0.221	0.257 ^a^	0.276 ^a^	0.184 ^b^	0.204 ^b^	0.174 ^b^	0.209 ^b^	0.058 ^b^	0.205 ^a^	0.251 ^a^	0.199 ^ab^	0.060 ^b^	0.190 ^b^
Furfural	1.044 ^a^	1.736	0.685 ^b^	1.58	0.702 ^b^	1.485	1.139 ^a^	0.708 ^a^	0.877 ^b^	0.527 ^b^	0.900 ^b^	0.511 ^b^	0.700	0.584	0.700	0.575	0.712	0.542
5-Methylfurfural	2.313 ^a^	2.978	1.811 ^b^	3.360	1.905 ^b^	2.743	2.610 ^a^	1.514 ^a^	2.183 ^b^	1.214 ^b^	2.060 ^b^	1.204 ^b^	1.213 ^b^	1.333	1.441 ^a^	1.291	1.297 ^b^	0.944
1-Methylpyrrole	0.213 ^a^	0.245	N.D ^b^	0.198	N.D ^b^	0.238	0.190 ^a^	0.377 ^a^	0.085 ^b^	0.252 ^b^	0.091 ^b^	0.251 ^b^	N.D ^b^	0.273	0.214 ^a^	0.267	N.D ^b^	0.229
1-Furfurylpyrrole	1.570	0.913	1.368	0.964	1.306	0.978	1.506 ^a^	0.727	1.005 ^b^	0.605	0.987 ^b^	0.638	0.669 ^b^	0.795 ^a^	1.154 ^a^	0.705 ^b^	0.685 ^b^	0.760 ^a^
Guaiacol	0.732	0.385	0.570	0.395	0.519	0.389	0.397 ^a^	0.409 ^a^	0.339 ^ab^	0.318 ^b^	0.274 ^b^	0.330 ^b^	0.079 ^b^	0.340	0.412 ^a^	0.337	0.064 ^b^	0.352
4-Ethylguaiacol	2.622 ^a^	1.127	2.062 ^b^	1.189	2.241 ^b^	1.159	1.644 ^a^	1.357 ^a^	1.381 ^b^	1.203 ^b^	1.422 ^b^	1.217 ^ab^	0.467 ^b^	1.155	1.421 ^a^	1.162	0.472 ^b^	1.219
2-Methoxy- 4-vinylphenol	3.971 ^a^	1.150	4.232 ^a^	1.079	4.741 ^b^	1.112	3.998 ^a^	1.379 ^a^	3.752 ^ab^	1.192 ^b^	3.431 ^b^	1.119 ^b^	1.529 ^b^	1.591	1.951 ^a^	1.514	1.472 ^b^	1.499

All values represented as the average of three replicates are the peak area ratio (peak area of each peak/peak area of internal standard). Different letters of the alphabet paired with a cardinal number within a particle size and extraction method indicate significant differences in the type of water according to Duncan’s test between each sample (*p* < 0.05). N.D: not detected.

**Table 4 foods-10-01347-t004:** Volatile compounds in coffee roasted at 245 °C for 17 min (model E-19 to E-27 for expresso, C-19 to C-27 for cold brew). The sample number was also assigned to the particle Scheme 355. μm: 19–21; 500 μm: 22–24; 710 μm: 25-27). The sample number assigned according to the size of the coffee bean was divided into three equal parts according to the type of water (filtered water: 19, 22, 25; tap water: 20, 23, 26; bottled water: 21, 24, 27).

Compounds	245 °C, 17 Min																	
	355 μm						500 μm						710 μm					
	Filtered Water		Tap Water		Bottled Water		Filtered Water		Tap Water		Bottled Water		Filtered Water		Tap Water		Bottled Water	
	E-19	C-19	E-20	C-20	E-21	C-21	E-22	C-22	E-23	C-23	E-24	C-24	E-25	C-25	E-26	C-26	E-27	C-27
2-Methylpyrazine	0.807	0.353	0.635	0.320	0.653	0.319	0.397 ^a^	0.589	0.350 ^b^	0.529	0.369 ^ab^	0.489	0.388	0.519 ^a^	0.643	0.451 ^b^	0.188	0.451 ^b^
2,3-Dimethylpyrazine	0.101	0.051	0.092	0.050	0.102	0.044	0.089	0.066	0.097	0.070	0.084	0.062	0.055	0.102 ^a^	0.057	0.090 ^a^	0.052	0.066 ^b^
2,5-Dimethylpyrazine	0.336	0.162	0.276	0.147	0.286	0.149	0.190	0.255	0.166	0.224	0.175	0.218	0.167 ^a^	0.238 ^a^	0.094 ^b^	0.212 ^b^	0.090 ^b^	0.198 ^b^
2,6-Dimethylpyrazine	0.396	0.203	0.322	0.193	0.333	0.190	0.229	0.305	0.207	0.287	0.216	0.271	0.198 ^a^	0.274 ^a^	0.105 ^b^	0.254 ^ab^	0.108 ^b^	0.246 ^b^
2-Ethylpyrazine	0.254	0.162	0.222	0.153	0.224	0.151	0.170	0.219	0.158	0.208	0.155	0.199	0.140 ^a^	0.210	0.079 ^b^	0.197	0.082 ^b^	0.179
2,6-Diethylpyrazine	0.148 ^a^	0.057	0.124 ^ab^	0.053	0.110 ^b^	0.054	0.070	0.062	0.040	0.056	0.059	0.057	0.021 ^b^	0.086 ^a^	0.029 ^a^	0.070 ^ab^	0.031 ^a^	0.064 ^b^
2-Ethyl- 3-methylpyrazine	0.353	0.195	0.324	0.184	0.303	0.188	0.212	0.251	0.203	0.244	0.226	0.225	0.175 ^a^	0.241	0.094 ^b^	0.229	0.101 ^b^	0.223
2-Ethyl- 5-methylpyrazine	0.199	0.112	0.167	0.113	0.168	0.116	0.111	0.132	0.107	0.128	0.118	0.128	0.095 ^a^	0.181	0.051 ^b^	0.166	0.061 ^b^	0.156
3-Ethyl- 2,5-imethylpyrazine	0.514 ^a^	0.325	0.387 ^ab^	0.320	0.243 ^b^	0.285	0.334 ^a^	0.388	0.240 ^b^	0.394	0.326 ^a^	0.363	0.193 ^a^	0.388	0.138 ^b^	0.389	0.140 ^b^	0.363
Isopropenyl pyrazine	0.322 ^a^	0.087	0.243 ^ab^	0.093	0.219 ^b^	0.095	N.D	0.115	0.24	0.093	N.D	0.102	0.102 ^a^	0.096 ^a^	0.020 ^b^	0.073 ^b^	0.024 ^b^	0.065 ^b^
2-Acetylfuran	0.788	0.302	0.638	0.310	0.651	0.287	0.415	0.485	0.368	0.435	0.381	0.417	0.254 ^a^	0.426 ^a^	0.158 ^b^	0.368 ^b^	0.161 ^b^	0.326^c^
2,2’-Bifuran	0.058	0.055	0.055	0.064	0.050	0.052	0.043	0.110	0.036	0.083	0.046	0.073	0.097 ^a^	0.069 ^a^	0.047 ^b^	0.047 ^b^	0.036 ^b^	0.040 ^b^
2,2’-Methylenebisfuran	0.904	0.297	0.816	0.319	0.814	0.287	0.676	0.359	0.627	0.378	0.679	0.359	0.625 ^a^	0.326	0.558 ^b^	0.320	0.498 ^b^	0.294
2,2’-[Oxybis (methylene)]bisfuran	1.376	0.932	1.364	0.974	1.365	0.933	1.508	0.867	1.430	0.944	1.718	0.952	1.172 ^a^	1.534	0.955 ^b^	1.735	0.959 ^b^	1.661
Furfuryl alcohol	1.973	0.775	1.646	0.857	1.872	0.888	1.254 ^ab^	1.198 ^a^	1.105 ^b^	1.032 ^b^	1.318 ^a^	0.970 ^b^	0.987 ^a^	1.717	0.734 ^b^	1.575	0.772 ^b^	1.616
Furfuryl acetate	2.977	1.540	2.633	1.536	2.386	1.492	2.096 ^a^	1.785	1.632 ^b^	1.760	2.016 ^a^	1.695	1.589 ^a^	2.280	1.017 ^b^	2.247	1.067 ^b^	2.061
Furfuryl propionate	0.242	0.158	0.228	0.153	0.192	0.142	0.204 ^a^	0.154	0.160 ^b^	0.164	0.202 ^a^	0.161	0.160 ^a^	0.196 ^a^	0.114 ^b^	0.184 ^ab^	0.108 ^b^	0.167 ^b^
Furfural	0.434	0.112	0.377	0.109	0.379	0.099	0.212	0.173	0.165	0.152	0.174	0.150	0.140	0.321 ^a^	0.138	0.252 ^b^	0.145	0.237 ^b^
5-Methylfurfural	0.448	0.234	0.402	0.198	0.416	0.210	0.182	0.347	0.179	0.312	0.214	0.327	0.255 ^a^	0.793	0.205 ^b^	0.748	0.194 ^b^	0.747
1-Methylpyrrole	N.D ^c^	0.297	0.459 ^a^	0.300	0.363 ^b^	0.306	0.272 ^a^	0.461	0.214 ^b^	0.434	0.244 ^ab^	0.416	0.272 ^a^	0.377	0.126 ^b^	0.343	0.110 ^b^	0.308
1-Furfurylpyrrole	0.915	0.544 ^b^	0.823	0.626 ^ab^	0.797	0.727 ^a^	1.293	0.600	1.123	0.649	1.378	0.591	0.811	0.640	0.819	0.555	0.834	0.555
Guaiacol	2.180	0.590	1.984	0.614	1.988	0.610	1.322	0.936	1.173	0.846	1.337	0.830	0.623 ^a^	0.567 ^a^	0.337^c^	0.510 ^ab^	0.439 ^b^	0.486 ^b^
4-Ethylguaiacol	4.936	2.002	4.545	2.037	4.498	1.945	4.220 ^a^	2.442	3.665 ^b^	2.424	4.255 ^a^	2.451	2.026 ^a^	1.944	1.381^c^	1.774	1.583 ^b^	1.835
2-Methoxy- 4-vinylphenol	1.736 ^a^	0.504 ^a^	1.562 ^b^	0.504 ^a^	1.689 ^ab^	0.413 ^b^	1.898 ^ab^	0.503	1.596 ^b^	0.510	2.018^s^	0.485	0.948^c^	1.192 ^a^	1.027 ^b^	0.923 ^b^	1.232 ^a^	0.877 ^b^

All values represented as the average of three replicates are the peak area ratio (peak area of each peak/peak area of internal standard). Different letters of the alphabet paired with a cardinal number within a particle size and extraction method indicate significant differences in the type of water according to Duncan’s test between each sample (*p* < 0.05). N.D: not detected.

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
