# Peer review of "Analysis of Volatile Compounds in Coffee Prepared by Various Brewing and Roasting Methods"

_foods, 2021, doi:10.3390/foods10061347_

Round 1
Reviewer 1 Report
The paper is very interesting and presents important data on the volatile composition of coffee. Some minor issues should be addressed:
1. Abstract: please list the particle sizes used in brewing
2. Page 1, lines 26-27: Please correct the sentence "The conventional process of coffee includes roasting, grinding, and extraction." It is unclear. Is it processing instead of process?
3. Introduction: Please add a part where the influence of grind size on volatile compounds is described. You mention that grind size influence will be investigated in this study, and therefore, a little part about that should be included in the Introduction.
4. Page 3, line 126: Why were 24 volatile compounds shown in Table 1 selected? What is their importance and the influence on aroma?
4. Table 1: Statistical data is missing (significant differences, SD or other errors of measurements)
5. Page 4, line 161: Fine grind 355 m?
6. Tables 2, 3 and 4: SD or other measurement errors are missing
7. Page 14, line 332: Shouldn't this section be called Conclusions?
Author Response
Reviewer 1
The paper is very interesting and presents important data on the volatile composition of coffee. Some minor issues should be addressed:
- Abstract: please list the particle sizes used in brewing
Answer: The particle sizes were added in abstract L14.
- Page 1, lines 26-27: Please correct the sentence "The conventional process of coffee includes roasting, grinding, and extraction." It is unclear. Is it processing instead of process?
Answer: Yes, the authors agree this point and now it was revised in L29.
- Introduction: Please add a part where the influence of grind size on volatile compounds is described. You mention that grind size influence will be investigated in this study, and therefore, a little part about that should be included in the Introduction.
Answer: The paragraph to state the importance of particle size was added with references in L35-37.
- Page 3, line 126: Why were 24 volatile compounds shown in Table 1 selected? What is their importance and the influence on aroma?
Answer: The selected volatiles are the main volatiles identified in coffee samples. In addition, these 24 volatiles were analyzed the most in our study and represent the aroma’s functional groups (chemical types) in coffee. It was revised in L132-135.
- Table 1: Statistical data is missing (significant differences, SD or other errors of measurements)
Answer: In Table 1, the values are retention time and RI of each volatiles. Thus there is no need to show statistical differences or errors.
- Page 4, line 161: Fine grind 355 m?
Answer: Now it was revised in L163-164.
- Tables 2, 3 and 4: SD or other measurement errors are missing
Answer: In Table 2-4 all data were treated by statistical tool and significant difference was shown in Tables. If we add SD for each values it would make the Tables look very confusing. When the editor or reviewer ask us we would submit all data as an appendix.
- Page 14, line 332: Shouldn't this section be called Conclusions?
Answer: It is true for this section to have both conclusions and discussion. However, more discussions were included in this section.
Reviewer 2 Report
The study covers an interesting field of coffee aroma analysis. The subject of the study in not particularly novel, but nevertheless it is a field of research where bringing together data and experiments that were performed separately in literature and taking another point of view is highly valuable.
The use of SPME and internal standard in slightly misleading as it is presented and should be corrected, please see the comments below.
13-14 The authors did not perform quantification with standards, therefore saying that the method showed something is not entirely correct. A vaguer wording is needed, for example “the results indicated”
16 Same as above, the authors did not quantify, therefor the proportion of Grind size was “to the measured signal of volatiles by GC/MS”, not the quantity.
16 Same for the relation to water, not the levels of compounds, but the measured GC/MS intensities. The use of word levels is appropriate, if it is clear that it is intensity levels, not concentration levels.
17 Please correct: significantly higher
20-22 Sentence unclear, please revise and explain more clearly what is first in this publication.
161 Unit missing
257-258 The importance of chloride ions in coffee brewing and sensory perception is questionable. Remove the chloride concentrations or provide reference why these could be important.
258-262 Inappropriate use of the reference [26]. These are some values from a study. If the authors don’t have the chlorine concentration of the actual waters that were used please remove this section. (also the actual values after boiling the water for preparing brew – chlorine will evaporate if water is boiled in a kettle, but could stay dissolved in an espresso boiler).
256-268 Discussion of other water parameters is missing: Sodium, Calcium and Magnesium content – cations play a role in the sensory perception of beverages. Carbonates neutralize acids and will cause different perception of acidity in coffee.
Overall: The first four points (13-14, 16,16,17) should be addressed overall in the manuscript. The measure of relative peak area for an analyte against an internal standard is not quantitatively sufficient to speak about concentrations. For the SPME extraction the matrix can have a dramatic effect on the sorption of the compounds and the only way to speak about concentrations is to perform quantitative analysis (isotope dilution analysis, standard addition, or multiple headspace injection analysis). As an example against this is that only changing the concentration of the brew (diluting it) will not affect all compounds the same way and will change the ratios of peak areas (while the real concentration ratio actually stayed the same). This is my experience working with similar types of SPME use. If authors have data that the coffee matrix does not effect in the same way the SPME fiber they are using the authors should provide method validation section that shows the robustness of the fiber to matrix effects.
Author Response
Reviewer 2
The study covers an interesting field of coffee aroma analysis. The subject of the study in not particularly novel, but nevertheless it is a field of research where bringing together data and experiments that were performed separately in literature and taking another point of view is highly valuable.
The use of SPME and internal standard in slightly misleading as it is presented and should be corrected, please see the comments below.
13-14 The authors did not perform quantification with standards, therefore saying that the method showed something is not entirely correct. A vaguer wording is needed, for example “the results indicated”
Answer: Yes, we, the authors agree this point and now it was revised in L16.
16 Same as above, the authors did not quantify, therefor the proportion of Grind size was “to the measured signal of volatiles by GC/MS”, not the quantity.
Answer: Yes, it was revised in L18-19.
16 Same for the relation to water, not the levels of compounds, but the measured GC/MS intensities. The use of word levels is appropriate, if it is clear that it is intensity levels, not concentration levels.
Answer: Yes, it was revised in L18-19.
17 Please correct: significantly higher
Answer: It was corrected as the reviewer’s comment in L20.
20-22 Sentence unclear, please revise and explain more clearly what is first in this publication.
Answer: We, the authors agree this point and now the sentence was removed.
161 Unit missing
Answer: The unit is μm and now it was clearly stated.
257-258 The importance of chloride ions in coffee brewing and sensory perception is questionable. Remove the chloride concentrations or provide reference why these could be important.
Answer: We, the authors totally agree the reviewer’s comment. Now the paragraph was removed.
258-262 Inappropriate use of the reference [26]. These are some values from a study. If the authors don’t have the chlorine concentration of the actual waters that were used please remove this section. (also the actual values after boiling the water for preparing brew – chlorine will evaporate if water is boiled in a kettle, but could stay dissolved in an espresso boiler).
Answer: We, the authors totally agree the reviewer’s comment. Now the paragraph was removed.
256-268 Discussion of other water parameters is missing: Sodium, Calcium and Magnesium content – cations play a role in the sensory perception of beverages. Carbonates neutralize acids and will cause different perception of acidity in coffee.
Answer: We, the authors totally agree the reviewer’s comment. Now the paragraph was removed.
Overall: The first four points (13-14, 16,16,17) should be addressed overall in the manuscript. The measure of relative peak area for an analyte against an internal standard is not quantitatively sufficient to speak about concentrations. For the SPME extraction the matrix can have a dramatic effect on the sorption of the compounds and the only way to speak about concentrations is to perform quantitative analysis (isotope dilution analysis, standard addition, or multiple headspace injection analysis). As an example against this is that only changing the concentration of the brew (diluting it) will not affect all compounds the same way and will change the ratios of peak areas (while the real concentration ratio actually stayed the same). This is my experience working with similar types of SPME use. If authors have data that the coffee matrix does not effect in the same way the SPME fiber they are using the authors should provide method validation section that shows the robustness of the fiber to matrix effects.
Answer: In the text we, the authors revised words as much as possible according to the reviewer’s recommendation such as L142, 145, 147, 166 etc…